# Tuberculosis treatment outcome: The case of women in Ethiopia and China, ten-years retrospective cohort study

Xiao Ma[1]*, Gebremeskel Mirutse[2], Alemayehu Bayray[2], Mingwang Fang[3]

**1** Department of Health-Related Social and Behavioral Science, West China School of Public Health, Sichuan University, Chengdu, China, **2** Schools of Public Health, College of Health Science, Mekelle University, Tigray, Ethiopia, **3** West China Hospital, Sichuan University, Chengdu, China

* 3301404937@qq.com

## Abstract

### Background

Every year tuberculosis kills above half million women all over the world. Nonetheless, the factor affecting TB treatment outcome of women was less frequently studied and compared among countries. Hence, this study was aimed to measure and compare outcome of treatment and the death size of these two countries.

### Method

Socio demographic and clinical data of women treated for all form of tuberculosis in the past ten years 2007–2016 were collected from total of eight hospitals and six treatment centers of Tigray and Zigong respectively. Then, we measured the magnitude of TB, level of treatment success and identify factors associated with the unsuccessful TB outcome.

### Result

In the past ten years, a total of 5603(41.5%) and 4527 (24.5%) tuberculosis cases were observed in Tigray and Zigong respectively. Of those with treatment outcome record a total of 2602(92%) in Tigray and 3916(96.7%) in Zigong were successfully treated. Total of 170 (6%) cases in Tigray and 36(0.8%) cases in Zigong were dead. In Tigray, retreatment cases (aOR, 0.29; 95% CI: 0.16–0.53) and MDR-TB cases (aOR, 0.31; 95% CI: 0.003, 0.27) were less likely to show treatment success. However,, HIV co-infected TB cases (aOR, 3.58; 95% CI: 2.47, 5.18) were more likely to show treatment success compared with unknown HIV status. In Zigong, women with MDR TB (aOR, 0.90; 95%CI: 0.24, 0.34) were less likely to show treatment success and women in the age category of 15–49 (aOR, 1.55; 95% CI: 1.08, 2.206) were more likely to show treatment success.

### Conclusion

Big number of tuberculosis cases and death were observed in Tigray comparing with Zigong. Hence, a relevant measure should be considered to improve treatment outcome of women in Tigray regional state.

**Data Availability Statement:** All relevant de-identified data are in the manuscript and its Supporting Information files.

**Funding:** The source of funding that support my work was from Mekelle University. The funder had no role in study design, data collection and analysis, decision to publish, or preparation of the manuscript.

**Competing interests:** The authors have declared that no competing interests exist.

**Abbreviations:** TB, Tuberculosis; HIV, Human immune deficiency Virus; MDR/TB, Multi drug resistance tuberculosis; CDC, The Centers for Disease Control and Prevention; WHO, The World Health Organization.

## Introduction

Despite the discovery of effective and affordable chemotherapy [1] tuberculosis kills 1.5 million people every year the death tall for women was 41.3% of the total death [2]. The gender difference in tuberculosis infection was not well understood. Nevertheless, TB kills more women annually than all the causes of maternal mortality combined [2]. In recent times, every year, at least 3.5 million women and children develop active TB among these 1.2 million cases died and more were left severely disabled [3–5].

Globally implementation of Direct observed treatment [DOTs] were saved 2.2 million of women and children [6] however there are enormous difference on the number of life saved and its factors affecting among regions. Few reports and studies attempts to compare and display the regional difference but the majorities were crude. For instance, WHO categorize Ethiopia and China among high TB burden countries [7–9]. But, its prevalence and treatment success reports were not specific for this group only inform about general population. Thus, the prevalence for Ethiopia was 192/100,000 and 67/100,000 was for China [10, 11] and the treatment success rate was 89% for Ethiopia and 94% for China [10]. In 2010, there was a regional report from high TB burden countries and out of 22 countries only 10 countries report contain specific data about women and children that time China notified a total of 869 092 TB cases out of this 17% were women and 0.8% were children [5] and Ethiopia notified a total of 150, 221 TB cases yet the size for women and children were not specified [5, 12].

Tigray which found in Northern part of Ethiopia [13] in 2015 notified a total 9,594 TB cases of both sex to a national TB control program. Accordingly 2,043 (21%) were smear positive with the cure rate of 74% (1,235) and 344 (4.2%) TB cases were died [14]. Zigong which is located in southeastern Sichuan and which is home of large number of TB cases [15, 16] in 2015 notified 1738 TB cases and among this 399(22%) were smear positive with the cure rate of 385 (96%) and among total 22(1.2%) cases were died.

Factors which identified as causes of unsuccessful treatment outcome for general population were retreatment cases, HIV co-infection, TB type and age in which repeatedly mentioned. However, this factors may not be found equally in all regions [17].

In conclusion, the evidence narrated in the above was not indicating the burden of TB in women specifically this implies that the existing study results and reports were crude. Thus, majority countries this time they lack specific proof which shows level of treatment outcomes and its factor affecting among these vulnerable group.

Declaring the above reasons, re-examining and comparing age and sex-aggregate data maintained by TB programs of these countries will be worth enough to look the profile, burden, treatment success and its factors affecting with in women. Moreover, finding of this study will help in tackling the limitation, shearing experience between countries and devise strategy to improve TB prevention and treatment program.

## Study settings and methods

### Tigray region

This study was conducted in Tigray (Ethiopia) and Zigong (China). Ethiopia is located in the Horn of Africa and is bordered by Kenya, Somalia, Sudan, Eritrea, and Djibouti. Administratively, Ethiopia is divided into nine regional states and two city administrative councils. The current population size was estimated 100 million[7, 18].Tigray is one of the nine national regional states of Ethiopia which is bordered by Eritrea in north and Sudan in the west. The region is administratively divided into seven Zones and 52 districts.

In 2010 among total population 2,441,158 (50.7%) were females with the total fertility rate of 5.1, agriculture is the main means of subsistence in the region in which 85% of the population lives in rural area.

In this region health care services are delivered through one specialized hospital, 15 general hospitals, 20 primary hospitals, 204 health centers, 712 health posts [village clinic] and 500 private health facilities. Then, in the region a total of 8,279 health professionals founds and 226 [2.7%] were doctors [14]. In Ethiopia Directly Observed Treatment (DOTS) was started in 1992 as a pilot and currently achieves 100 percent geographical coverage and recently 92% of public hospitals and health centers offer DOTS [19].The role of health facility in TB prevention and control was not centralized that means all hospitals and majority of health centers which have the diagnostic technology are allowed to diagnosis TB and providing DOTs service[17, 19, 20]. Though, the role of health posts (village clinics) was limited only they provide health education, refer TB suspects for investigation and collect sputum smears, retrieve absentees/ defaulters and in few place they can provide DOTs for case who is very far from health institution[21].

## Zigong region

China's National TB control Program started to implement the international recommended directly observed treatment, short-course (DOTS) strategy in 1991, and expanded the DOTS program to the entire country by 2005 [8]. Zigong is found in south-west China Sichuan province and currently this county has a total population 3.28 million. The existing health care system was organized into a three-tier health care delivery system and tuberculosis control program is centralized. Thus, the basic unit of TB health care is the specialized County TB dispensary (CTD) with the responsibilities of TB diagnosis, treatment and patient management guided by the National TB control program[11].Whereas, the non-CTD's role in TB control program is to refer suspected TB patients to CTDs [22] all patient took anti TB drug three times per week [8] and DOTs observers get paid 60 Yuan (US\$1 $\approx$ CNY7) per TB patient for the standardized treatment regimen of $\geq$6 months [23] The recommended treatment regimen for new TB case and retreated TB cases were the same 2HRZS/4HR and 2HRZSE/6HRE [16]

## Study design, sampling technique and data collection

Using retrospective cross-sectional study design we reviewed all form of TB cases of women treated in the years of January 2007 to December 2016 in both countries. In Tigray DOTs was not centralized. Hence, all health facilities are allowed to diagnosis TB and provide DOTs. In Zigong DOTs services were centralized and it is provided only in specific government assigned health facility. In Tigray TB patient information was found in log book registered manually whereas in Zigong it was found in digital and manual form. Hence, considering the logistic constraint in Tigray among 16 hospitals eight hospitals that provide DOTs service for ten years and above were randomly selected then trained data collectors and supervisors were assigned to collect the information. Nonetheless, in Zigong there were only six TB treatment centers and information about TB patients were found in digital form. Thus, all data's was extracted from the Excel sheet. Since, sample was fully anonym zed before we accessed and patient informed consent was waved.

## Data analysis

After checking the completeness, data was entered and analyzed using SPSS Version 21. Then, descriptive analysis such as frequency, mean and standard deviation were computed and compared. Additionally, binary logistic regression was executed to examine the association of

independent variable with unsuccessful treatment outcomes. Hence, P-value less than of 0.05 was used as significant value. Finally, variables significant in binary logistic regression were analyzed again using multiple logistic regressions to identify variables which augment unsuccessful treatment outcomes in women and children.

### Ethical clearance

The study was passed through the ethical approval procedure of Sichuan University College of public health Chengdu, China and C152 HS/IRB of Mekelle University Ethiopia.

## Results

### Socio demographic and clinical characteristic of women

The past ten years (January 2007- December2016) in Tigray a total of 13,435 and in Zigong 18,423 TB cases were identified. Among this 5603(41.7%) cases in Tigray and 4527(24.5%) in Zigong were women in the age category of 15–49. The mean age of case in Tigray was 36 ±15 years and in Zigong was 44 ±17years. Looking the age category in Tigray 4274 (76.3) and Zigong 2798(61.8) of women were in the age category of 15–49.

In addition among all TB cases 1175(21%) in Tigray and 2151(47.5%) in Zigong were pulmonary positive cases. Then, 21(0.4%) in Tigray and 16 (0.3%) in Zigong were MDR-TB. Also, 1048(18.7%) TB/HIV cases were identified in Tigray whereas no HIV documentations were found in Zigong. (Table 1)

### Women clinical character and tuberculosis treatment in Tigray and Zigong

Over the study period, total of 5603(41.7%) and 4527(24.5%) tuberculosis cases were registered in Tigray and Zigong respectively. Then, in Tigray among all cases 2728(48.7%) TB cases were

**Table 1. General characteristic of women treated for tuberculosis in Tigray and Zigong January 2007–December 2016[N = 5603 and N = 4527].**

| Characteristics | Category | Countries | |
|---|---|---|---|
| | | Tigray | Zigong |
| | | N (%) | N (%) |
| Age category | | | |
| | 15–49 | 4274 (76.3) | 2798(61.8) |
| | ≥ 50 | 1329(23.7) | 1729(38.2) |
| TB type | Smear Positive | 1175(21) | 2151(47.5) |
| | Smear Negative | 2068(36.9) | 2362(52.2) |
| | Extra Pulmonary | 2360(42.1) | 14(0.3) |
| Patient category | New | 5265(94) | 4260(94.1) |
| | Relapse | 162(2.9) | 266(5.9) |
| | Defaulter/failure | 12(0.2) | 0 |
| | Unrecorded | 164(0.6) | 2 |
| MDR-TB | MDR | 21(0.4) | 16(0.3) |
| | NMDR | 5582(99.6) | 4511(99.7) |
| HIV Status | Positive | 1048 (18.7) | NA |
| | Negative | 3728(66.5) | NA |
| | Un known | 827(14.8) | NA |
| Year | 2007–11 | 3339(60) | 2432(53.7) |
| | 2012–16 | 2264(40) | 2095(46.3) |
| | Total | 5603 | 4527 |

transferred to their nearby health facilities, 71(1.3%) have no record of their treatment outcome and in Zigong 480 (10.6%) case their treatment outcome were not recorded. Therefore, transfer out and cases with unknown treatment outcome were not included in the analysis. So, a total of 2804(50%) cases from Tigray and 4047(89%) from Zigong were involved in the analysis. Accordingly, 2602(92%) in Tigray and 3916(96.7%) case in Zigong were successfully treated. The cure rate of pulmonary positive cases out of 528 cases 477[90%] in Tigray where as in Zigong among 1891cases 1801 [95%] were cured. (Table 2)

## The trend of treatment success in women

The past ten years trend of treatment success was assessed the percentage of treatment success in Tigray was between 86%-98% and for Zigong was between 94%-98%. In Tigray the lowest treatment success was seen in the year 2007 which is 81% and in Zigong the lowest treatment success was seen in 2013 and its percentage was 94%. The average treatment success for Tigray was 92% where as in Zigong it was 96.6 percent.

The graph for treatment success indicates in Tigray the past ten years there was constant increment then sharp decrease in the year 2016. But in Zigong it was a constant increment. (Fig 1)

## Trend of tuberculosis death in Tigray and Zigong

In the past ten years a total of 170 [6%] case in Tigray and 36 [0.9%] cases in Zigong were reported died which is 6:1 ratio. Besides, in Tigray the peak death was seen in the year 2007 [12%] and in Zigong were in 2011 (0.57%). looking the trend of death in Zigong it was constant with the average death of 0.8 per year where as in Tigray it was a decreasing pattern and the average death per year was 5.8 per 100 cases. (Fig 2)

**Table 2. Women clinical factors and level of unsuccessful treatment outcome in Tigray and Zigong from January 2007–December 2016[N = 2804 and N = 4047].**

| Characteristic | Country | | | | | |
|---|---|---|---|---|---|---|
| | Tigrai | | | Zigong | | |
| | Total | Unsuccessful N (%) | Death | Total | Unsuccessful N (%) | Death |
| **Type TB** | | | | | | |
| P-Positive | 528 | 51(9.6) | 35(6.6) | 1889 | 60(3.2) | 22(1.2) |
| P-Negative | 1062 | 83(7.8) | 73(6.8) | 2156 | 70(3.2) | 13(0.6) |
| E- pulmonary | 1214 | 71(5.8) | 62(5.1) | 2 | 1(50) | 1(50) |
| **Treatment** | | | | | | |
| New | 2709 | 178(6.5) | 158(5.8) | 3822 | 122(1.7) | 32(0.83) |
| Re treatment | 95 | 24(25.2) | 12(12.6) | 225 | 9(4.0) | 4(1.7) |
| **Age** | | | | | | |
| 15–49 | 2155 | 156(7.2) | 134(6.1) | 2494 | 69(2.8) | 11(0.4) |
| ≥50 | 649 | 46(6.9) | 36(5.4) | 1553 | 62(4.1) | 25(1.6) |
| **Drug resistance** | | | | | | |
| MDR | 7 | 6(75) | 3(42.8) | 13 | 3(31.2) | 0 |
| NMDR | 2797 | 196(7.0) | 167(5.9) | 4034 | 128(3.2) | 36(0.9] |
| **HIV status** | | | | | | |
| Positive | 596 | 60(10) | 56(9.3) | 0 | NA | NA |
| Negative | 1838 | 88(4.7) | 70(3.8) | 4047 | NA | NA |
| Not tested | 370 | 54(14.5) | 49(13.2) | 0 | NA | NA |
| **Total** | 2804 | 202 | 170 | 4049 | 137 | 36 |

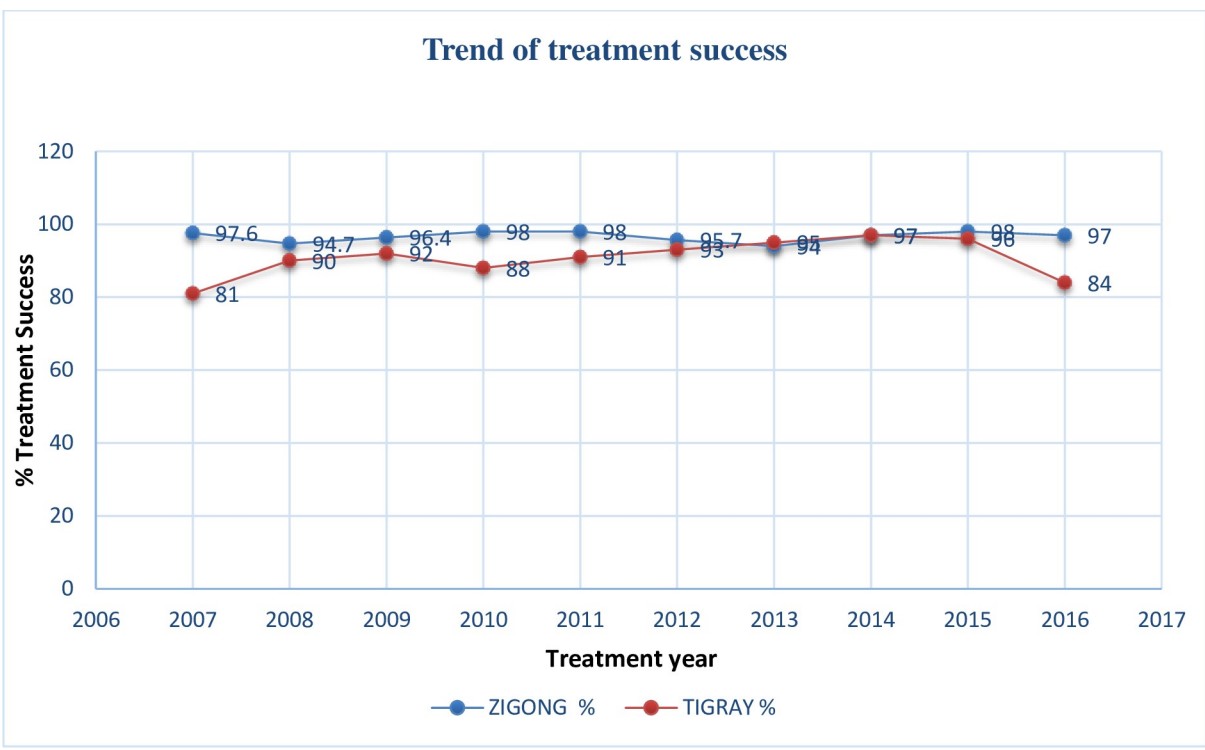

**Fig 1. Trend of women treatment success in Tigray and Zigong January 2007–December 2016 N = 2084 and N = 4047.**

## BLR: Sociodemographic and clinical factors associated with unsuccessful treatment outcomes in women

In this study, we did bivariate logistic regression to identified factors that have association with unsuccessful treatment outcomes for both countries.

Accordingly, in Tigray those TB cases in retreatment category MDR cases, pulmonary positive and negative were more likely to show unsuccessful treatment outcome comparing with their counter parts and TB/ HIV co infected and HIV negative cases have more likely to have treatment successes comparing with those unknown HIV status.

In Zigong MDR cases and age between 15–49 TB have more likely to have unsuccessful treatment outcome compared with others. (Table 3)

## MLR: Factors associated with treatment outcome of women in Tigray and Zigong

Factors significant at P-value < 0.05 in the bivariate logistic regression were took and analyzed again in multivariate regression to identify the predictor variables. Then, in Tigray treatment success was less likely for women who were categorized as retreatment (adjusted OR, 0.29; 95% CI: 0.16–0.53) compared to new cases, women with multi drug resistant (adjusted OR, 0.31; 95% CI: 0.003, 0.27) compared with non-drug resistant. But, HIV co infected TB cases were 1.59 times more likely to have treatment success (95% CI: 2.47, 5.18) compared with Unknown HIV status.

In Zigong, women in the age category of 15–49 years have 1.55 more likely to show treatment success (95% CI: 1.08, 2.206) compared with older age. But, women with MDR TB were

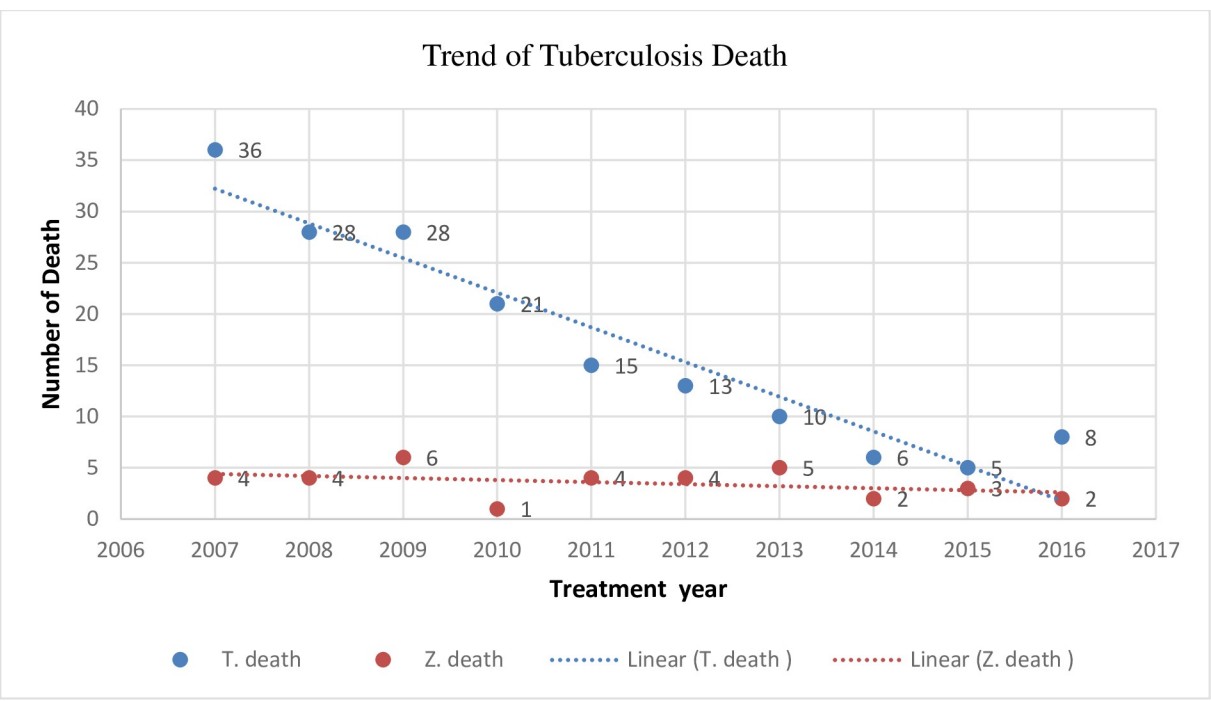

**Fig 2. Trend of tuberculosis death in the past ten years (January 2007- December 2016) in women, Tigray N = 2084 and Zigong N = 4047.**

less likely to show treatment success (Adjusted OR, 0.90; 95%CI: 0.24, 0.34) compared with non-drug resistance cases. (Table 4)

## Discussion

Tuberculosis is exacerbated by malnutrition and frequently affects economically active young adults [24, 25]. Thus, women of reproductive age are more likely to develop active TB if they encounter TB bacteria and they are less likely to seek help for TB symptoms than men [4]. In this study the mean age of TB case in Tigray was 36 years with SD±15 and in Zigong was 44years with SD±17. Tigray finding was similar with study done in sidama and Gojjam for general population [17, 20]. But, comparing with Zigong more younger women were get infected in Tigray than Zigong and this could be; the age distributions tuberculosis in Africa has been severely skewed by the human immunodeficiency virus epidemic [26], under nutrition was a major public health problems in Tigray and it has considerable effect in provoking tuberculosis infection [27, 28] and the demographic structural difference between this two country may be other cause large number of old peoples were found in Zigong than Tigray.

In 2014, globally TB killed 480,000 women [2]. Correspondingly, in this retrospective study a total of 170 [6%] women in Tigray and 36 [0.8%] in Zigong were died. The death toll in Tigray was higher than the annual report of the Regional Health Bureau for both sex 3.6% [29], retrospective study done in Gojjam for both sex 3.7% and global report of TB death for women and children 4.2% [10, 14, 17]. In cases of Zigong it is similar with the global report for general population [10]. Comparing with Zigong, more death occurred in Tigray. This difference could be the prevalence of TB/HIV co morbidity was high in sub-Saharan country[30], poor health care seeking behavior of women in the region which leads to diagnostic delay [31] and poor treatment adherence in high TB burden countries[32] were in favor of high number of deaths in Tigray.

**Table 3. MLR factors associated with treatment outcome of women in Tigray and Zigong January 2007 December 2016 N = 2084 and N = 4047.**

| Characteristic | Tigray | | | | | Zigong | | | | |
|---|---|---|---|---|---|---|---|---|---|---|
| | Total | Unsuccessful N% | β | COR(95%CI) | P value | Total | Unsuccessful N% | β | COR(95%CI) | P value |
| **Type of TB** | | | | | | | | | | |
| **P-Positive** | 528 | 51(9.6) | -0.59 | 0.55(0.38,0.81) | 0.002 | 1889 | 60(3.2) | 0.23 | 1.023(0.72,1.45) | 0.899 |
| **P-Negative** | 1062 | 83(7.8) | -0.36 | 0.7(0.50,0.97) | 0.035 | 2156 | 70(3.2) | -3.34 | 0.03(0.002,0.54) | 0.017 |
| **E. pulmonary** | 1214 | 71(5.8) | | 1 | 1 | 2 | 1(50) | | 1 | 1 |
| **Treatment** | | | | | | | | | | |
| **Re-treatment** | 95 | 24(25.2) | -1.57 | 0.28(0.13,0.34) | 0.0001 | 3822 | 122(1.7) | 0.23 | 0.82(0.41,1.62) | 0.561 |
| **New** | 2709 | 178(6.5) | | 1 | 1 | 225 | 9(4.0) | | 1 | 1 |
| **Age** | | | | | | | | | | |
| **15–49** | 2155 | 156(7.2) | -0.23 | 0.98(0.69,1.37) | 0.89 | 2494 | 69(2.8) | 0.37 | 1.46(1.03,2.07) | 0.033 |
| **≥ 50** | 649 | 46(6.9) | | 1 | 1 | 1553 | 62(4.1) | | 1 | 1 |
| **Drug resistance** | | | | | | | | | | |
| **MDR** | 7 | 6(75) | -4.37 | 0.013(0.002,0.11) | 0.001 | 13 | 3(31.2) | -2.214 | 0.109(0.03, 0.42) | 0.001 |
| **NMDR** | 2797 | 196(7.0) | | 1 | 1 | 4034 | 128(3.2) | | 1 | 1 |
| **HIV status** | | | | | | | | | | |
| **Negative** | 1838 | 88(4.7) | 1.223 | 3.39(2.37,4.87) | 0.001 | ND | ND | | ND | ND |
| **Positive** | 596 | 60(10) | 0.423 | 1.53(1.03,2.26) | 0.035 | ND | ND | | ND | ND |
| **Not tested** | 370 | 54(14.5) | | 1 | 1 | ND | ND | | ND | ND |
| **Total** | 2804 | 202 | | | | 4047 | 131 | | | |

Treatment success is sum of patients cured and those who have completed treatment. Hence, patient compliance is a key factor in treatment success [10]. In this study, the overall treatment success rate of all TB cases was 92% in Tigray and 96.6% in Zigong. The finding of Tigray was similar with studies done in west Gojjam, Sidama and Addis Ababa 91.5% [17, 20,

**Table 4. Multiple logistic regression factors affecting treatment outcome of women in Tigray and Zigong January 2007 December 2016 N = 2084 and N = 4047.**

| Characteristic | Tigray | | | | | Zigong | | | | |
|---|---|---|---|---|---|---|---|---|---|---|
| | Total | Un success N% | COR(95%CI) | AOR(95% CI) | P-value | Total | Un success N% | COR(95%CI) | AOR(95% CI) | P-value |
| **Type of TB** | | | | | | | | | | |
| P-Positive | 528 | 51(9.6) | 0.55(0.38,0.81) | 0.89(0.58,1.37) | 0.61 | 1889 | 60(3.2) | 1.023(0.72,1.45) | 0.91(0.63,1.31) | 0.614 |
| P-Negative | 1062 | 83(7.8) | 0.7(0.50,0.97) | 0.78(0.56,1.09) | 0.15 | 2156 | 70(3.2) | 0.03(0.002,0.54) | 0.025(0.002,0.41) | 0.01 |
| E. pulmonary | 1214 | 71(5.8) | 1 | 1 | | 2 | 1(50) | 1 | 1 | |
| **T. Category** | | | | | | | | | | |
| Re-treatment | 95 | 24(25.2) | 0.28(0.13,0.34) | 0.29(0.16–0.53) | 0.0001 | 3822 | 122(1.7) | 0.82(0.41,1.62) | 1.12(0.53,2.35) | 0.76 |
| New | 2709 | 178(6.5) | 1 | 1 | | 225 | 9(4.0) | 1 | 1 | |
| **Age** | | | | | | | | | | |
| 15–49 | 2155 | 156(7.2) | 0.98(0.69,1.37) | 1.17(0.81,1.69) | 0.378 | 2494 | 69(2.8) | 1.46(1.03,2.07) | 1.55(1.08,2.206) | 0.016 |
| ≥ 50 | 649 | 46(6.9) | 1 | 1 | | 1553 | 62(4.1) | 1 | 1 | |
| **Drug resistance** | | | | | | | | | | |
| MDR | 7 | 6(75) | 0.013(0.002,0.11) | 0.31(0.003,0.27) | 0.002 | 13 | 3(31.2) | 0.109(0.03, 0.42) | 0.090 (0.024,0.34) | 0.0001 |
| NMDR | 2797 | 196(7.0) | 1 | 1 | | 4034 | 128(3.2) | 1 | 1 | |
| **HIV status** | | | | | | | | | | |
| Negative | 1838 | 88(4.7) | 3.39(2.37,4.87) | 3.58(2.47,5.18) | 0.0001 | ND | ND | ND | ND | |
| Positive | 596 | 60(10) | 1.53(1.03,2.26) | 1.59(1.07,2.38) | 0.022 | ND | ND | ND | ND | |
| Not tested | 370 | 54(14.5) | 1 | 1 | | ND | ND | ND | ND | |
| **Total** | 2804 | 202 | | | | 4047 | 131 | | | |

33] the success rate for both sex. But, higher than the WHO report 89% for both sex [10] and the success of Zigong is higher than [10, 34, 35]. Thus, the better treatment success in both countries could be since these studies assess only women and this group have good treatment adherence as a result the percentage of treatment success was better comparing with general population.

A patient is considered "cured" when sputum smear examination is bacteriologically negative in the last month of treatment and on at least one previous occasion[17].The cure rate for pulmonary positive cases was 90% in Tigray and 95% in Zigong. So, the finding of Tigray was slightly higher than 2015/16 annual report of the region and study done in west Gojjam [14, 17, 29] for general population. But, in case of Zigong it is similar with the WHO global report for Chinese population [10]. Therefore, the reason of less cure rate in Tigray comparing with Zigong could be; the presence of large number of TB/HIV co infected women in the region, high under nutrition level of the region and poor consistency sputum test result of the region were made Tigray region to show less cure rate than Zigong.

Retreatment case is patient who has been treated for one month or more with anti-TB drugs in the past[2] and many studies indicate that re treatment case have high chance of poor treatment outcome or failure rate. Similarly, in our study re-treatment cases were 79% less likely (adjusted OR, 0.29; 95% CI: 0.16–0.53) to have treatment success compared to new cases and this finding was parallel with the study done in Gojjam, Sidama, Tigray and Uganda [13, 17, 20, 36]. Yet, in Zigong it was not significant. The less treatment successes seen in the retreatment cases in Tigray could be the presence of high number of multidrug resistance TB in retreatment cases and late introduction of drug sensitivity test in the region made the cases to take anti TB drug without knowing there MDR status.

Globally, an estimated 3.3% of new TB cases and 20% of previously treated cases have MDR-TB [2]. In Ethiopia 2.7% of the new and 14% of the previously treated TB cases expected to have had rifampicin or multi drug resistant TB [10]. Report about MDR-TB started in 2011 in both countries. Then in Tigray total of 28(0.4%) MDR-TB cases were reported and majorities were identified in 2016 which is 16(3.7%) cases. Where as in Zigong 16 (0.3%) MDR-TB cases were identified.

According to global TB report only 50% of MDR-TB patients were successfully treated[10]. But in this study 6(85.7%) women out of 7 MDR TB cases in Tigray and 3 (23%) women out of 13 MDR TB cases in Zigong were not successfully treated. this is consistent with study done in India China and Ethiopia [37, 38]. In both study area patients with drug resistant tuberculosis were less likely to have treatment success. In Tigray 69% less-likely (Adjusted OR, 0.31; 95% CI: 0.003, 0.27 and in Zigong 9% less likely (adjusted OR, 0.09; 95% CI: 0.024, 0.34 successful treatment outcome compared with non-drug resistant. But, comparing with Zigong high percentage of women in Tigray were less likely to show treatment success this could be; in Tigray MDR TB diagnoses took long time because the diagnostic center was found only in the capital city, MDR TB patient remain without treatment for long time until the result arrive and the long-term treatment ended many women with adverse effect this all might made the outcome poor and many studies support this finding [25, 39].

## Conclusions

Evidence presented in this study shows tuberculosis was one of the major public health concerns for women in both countries. However, poor level of treatment success and high mortality were seen in Tigray compared with Zigong. Besides, factors boost unsuccessful treatment outcome were many in case of Tigray than in Zigong. Hence, national health policy makers of Ethiopia and Tigray regional health office should give due attention to this specific group.

## Supporting information

**S1 Data Set.**
(SAV)

**S2 Data Set.**
(SAV)

## Acknowledgments

The authors would like to thank Sichuan University and Mekelle University for guiding me. We also would like to thank Zigong CDC, Tigray Regional Health Bureau and TB treatment centers of each country which allow us to use the data. We would like also to extend our thanks to all the data collectors.

## Author Contributions

**Conceptualization:** Xiao Ma, Gebremeskel Mirutse.

**Formal analysis:** Gebremeskel Mirutse.

**Writing – original draft:** Gebremeskel Mirutse.

**Writing – review & editing:** Xiao Ma, Alemayehu Bayray, Mingwang Fang.

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
