## [Decision Letter · Decision Letter 0]

31 Jul 2019

PONE-D-19-17086

Tuberculosis treatment outcome: The case of women in Ethiopia and China, Ten-Years Retrospective Cohort study

PLOS ONE

Dear Dr. Xiao Ma,

Thank you for submitting your manuscript to PLOS ONE. After careful consideration, we feel that it has merit but does not fully meet PLOS ONE’s publication criteria as it currently stands. Therefore, we invite you to submit a revised version of the manuscript that addresses the points raised during the review process.

We would appreciate receiving your revised manuscript by Sep 14 2019 11:59PM. To enhance the reproducibility of your results, we recommend that if applicable you deposit your laboratory protocols in protocols.io, where a protocol can be assigned its own identifier (DOI) such that it can be cited independently in the future. For instructions see: http://journals.plos.org/plosone/s/submission-guidelines#loc-laboratory-protocols

We look forward to receiving your revised manuscript.

Kind regards,

HASNAIN SEYED EHTESHAM

Academic Editor

PLOS ONE

2. In ethics statement in the manuscript and in the online submission form, please provide additional information about the patient records/samples used in your retrospective study. Specifically, please ensure that you have discussed whether all data/samples were fully anonymized before you accessed them and/or whether the IRB or ethics committee waived the requirement for informed consent. If patients provided informed written consent to have data/samples from their medical records used in research, please include this information.

"The authors would like to thank Sichuan University and Mekelle University for funding this study."

Please remove any funding-related text from the manuscript and let us know how you would like to update your Funding Statement. Currently, your Funding Statement reads as follows: 'NO'

Additional Editor Comments (if provided):

I recommend this manuscript for Major revision.

Reviewers' comments:

Reviewer's Responses to Questions

**Comments to the Author**

1. Is the manuscript technically sound, and do the data support the conclusions?

Reviewer #1: No

Reviewer #2: Yes

2. Has the statistical analysis been performed appropriately and rigorously? 

Reviewer #1: Yes

Reviewer #2: I Don't Know

3. Have the authors made all data underlying the findings in their manuscript fully available?

Reviewer #1: No

Reviewer #2: Yes

4. Is the manuscript presented in an intelligible fashion and written in standard English?

Reviewer #1: No

Reviewer #2: Yes

5. Review Comments to the Author

Reviewer #1: Retrospective analysis of cohorts in two countries were made and data was analyzed. Following points were highlighted during the review:

1. Significant number of tuberculosis cases and death were observed in Tigray than Zigong region. Highlighting the shortcomings in implementation of TB programme in northern part of ethiopia.

2. Why, No data is available for TB-HIV co-morbid condition in Zigong region.

3. Factors responsible to unsuccessful treatment outcome in Tigray needs to be highlighted at large.

4. Factors responsible for less treatment success in Re-treatment cases needs to be highlighted also.

5. In Line 75,76 stating the lacunae of the study outcomes; Thus, globally this time we lack specific proof which shows level of treatment outcomes and its factor affecting in these vulnerable group.

6. Why only these two regions of these two countries are chosen for comparison, as the population, and other demographic conditions of these two countries are totally different. Reason needs to be provided.

However, such studies needs to be planned to highlight the lacunae in implementation of the national TB control programme and to study successful treatment outcomes.

Reviewer #2: 1. Whether the study population was native to the said region or migrant population also there needs to be clarified

2. Since the occupation of people from Tigray is mentioned. Same needs to be highlighted about the study subjects from Zigong. Also of any corelation is deduced from the same.

3. Any positive family history of TB in study subjects should also been recorded.

4. It is mentioned that retreatment or MDR cases showed poor treatment success as compared to those with HIV coinfection needs to be elaborated upon.

6. PLOS authors have the option to publish the peer review history of their article (what does this mean?). If published, this will include your full peer review and any attached files.

Reviewer #1: No

Reviewer #2: No

---

## [Author Response · Author response to Decision Letter 0]

4 Oct 2019

Review Comments to the Author

Reviewer #1: Retrospective analysis of cohorts in two countries were made and data was analyzed. Following points were highlighted during the review:

Author’s appreciation 

We would like to thank the reviewer for careful and thorough reading of this manuscript and for the thoughtful comments and constructive suggestions, which help to improve the quality of this manuscript. Our response follows (the reviewer’s comments are in non-italic form). 

1. Significant number of tuberculosis cases and death were observed in Tigray than Zigong region. Highlighting the shortcomings in implementation of TB program in northern part of Ethiopia 

Dear reviewer: I add reasons to explain the short comings in implementation of TB program a Tigray region in each discussion. 

2. Why, No data is available for TB-HIV co-morbid condition in Zigong region.

Dear reviewer: There was a separate registration form for TB-HIV co-morbid which was not allowed to access. I did many attempts with my professor to have the information but it was not possible. 

3. Factors responsible to unsuccessful treatment outcome in Tigray need to be highlighted at large.

Dear reviewer: I highlighted many factors responsible to unsuccessful treatment outcome in the discussion part also

4. Factors responsible for less treatment success in Re-treatment cases needs to be highlighted also,

Dear reviewer: the reason I highlighted is “The less treatment successes seen in the retreatment cases in Tigray could be the presence of high number of multidrug resistance TB in retreatment cases and with late introduction of drug sensitivity test in the region made the cases to take anti TB drug without knowing there MDR status. 

5. In Line 75,76 stating the lacunae of the study outcomes; Thus, globally this time we lack specific proof which shows level of treatment outcomes and its factor affecting in these vulnerable group:

Dear reviewer: Many of the study compare men treatment outcome with women. But, there are very few studies which try to compare the treatment outcome among women. Hence, based on this I considered as we luck specific proof which show the level of treatment outcome 

6. Why only these two regions of these two countries are chosen for comparison, as the population, and other demographic conditions of these two countries are totally different. Reason needs to be provided. However, such study needs to be planned to highlight the lacunae in implementation of the national TB control programmer and to study successful treatment outcomes.

Dear reviewer: The main reason to chooses this regions were ,Tigray region which is located in northern part of Ethiopia has best performance in TB prevention and treatment program and Zigong has the least performance in TB prevention and treatment and little rural among Sichuan provinces. Therefore, select these regions will show as comparable level of treatment outcome of the country. 

Reviewer #2: 1. Whether the study population was native to the said region or migrant population also there needs to be clarified

Dear reviewer: This study did not consider the cases weather it is migrant or non-migrant because this information was not found in the TB registration logbook of Tigray region where as in Zigong there is but the number of migrant cases is not as significant. 

2. since the occupation of people from Tigray is mentioned. Same needs to be highlighted about the study subjects from Zigong. Also of any correlation is deduced from the same.

Dear reviewer: I did not mention any occupation in Tigray 

3. Any positive family history of TB in study subjects should also been recorded.

Dear reviewer: Yes, it was good to record any positive family history. But, both registrations did not indicate any positive family history. Besides, there is column which indicates number of contacts person which used to trace contact weather they are infected or not. 

4. It is mentioned that retreatment or MDR cases showed poor treatment success as compared to those with HIV confection needs to be elaborated upon.

Dear reviewer: Here is the sentence on line 257-259 which says , HIV co infected TB cases were 1.59 times more likely to have treatment success (95% CI: 2.47, 5.18) compared with Unknown HIV status. In the result part and the reason behind was while The TB/ HIV prevention and treatment guideline indicates that any TB case should be tested for HIV and linked to Pre-ART treatment. But, some TB patent reject the HIV testing offer and delay themselves to take correct treatment protocol as well some health facility forgot to offer the test and delay in linking to ART program this made them to show poor treatment outcome than those known HIV positives 

6. PLOS authors have the option to publish the peer review history of their article (what does this mean?). If published, this will include your full peer review and any attached files.

Do you want your identity to be public for this peer review? For information about this choice, including consent withdrawal, please see our Privacy Policy.

Reviewer #1: No

Reviewer #2: No

---

## [Editor Report · Decision Letter 1]

15 Oct 2019

Tuberculosis treatment outcome: The case of women in Ethiopia and China, Ten-Years Retrospective Cohort study

PONE-D-19-17086R1

Dear Dr. Xiao Ma,

We are pleased to inform you that your manuscript has been judged scientifically suitable for publication and will be formally accepted for publication once it complies with all outstanding technical requirements.

With kind regards,

HASNAIN SEYED EHTESHAM

Academic Editor

PLOS ONE

Additional Editor Comments (optional):

The Authors have modified the manuscript seeking in mind the comments of the Reviewers. Reviewer 1 have No major comments and others have been taken care of. Reviewer 2 has also asked some questions and sought clarifications and elaborations on some points and these have been provided by the Authors. I believe this manuscript attempts to address treatment outcome in high disease burden country.
---

## [Editor Report · Acceptance letter]

7 Nov 2019

PONE-D-19-17086R1 

Tuberculosis treatment outcome: The case of women in Ethiopia and China, Ten-Years Retrospective Cohort study 

Dear Dr. Ma:

I am pleased to inform you that your manuscript has been deemed suitable for publication in PLOS ONE. Congratulations! Your manuscript is now with our production department. 

With kind regards,

on behalf of

Prof HASNAIN SEYED EHTESHAM 

Academic Editor

PLOS ONE